Perspective  

**Subject Area:**
biochemistry/biotechnology/molecular biology

lignin, secondary cell wall, metabolic channelling, membrane transport, oxidative polymerization, reactive oxygen species

**Author for correspondence:**
Richard A. Dixon
e-mail: richard.dixon@tamu.edu

†An invited perspective to mark the election of the author to the fellowship of the Royal Society in 2018.

# Lignin biosynthesis: old roads revisited and new roads explored

Richard A. Dixon[1,2,†] and Jaime Barros[2]

[1]Hagler Institute for Advanced Studies and Department of Biological Sciences, Texas A&M University, College Station, TX, USA
[2]BioDiscovery Institute and Department of Biological Sciences, University of North Texas, 1155 Union Circle #311428, Denton, TX 76203-5017, USA

(iD) RAD, 0000-0001-8393-9408

Lignin is a major component of secondarily thickened plant cell walls and is considered to be the second most abundant biopolymer on the planet. At one point believed to be the product of a highly controlled polymerization procedure involving just three potential monomeric components (monolignols), it is becoming increasingly clear that the composition of lignin is quite flexible. Furthermore, the biosynthetic pathways to the major monolignols also appear to exhibit flexibility, particularly as regards the early reactions leading to the formation of caffeic acid from coumaric acid. The operation of parallel pathways to caffeic acid occurring at the level of shikimate esters or free acids may help provide robustness to the pathway under different physiological conditions. Several features of the pathway also appear to link monolignol biosynthesis to both generation and detoxification of hydrogen peroxide, one of the oxidants responsible for creating monolignol radicals for polymerization in the apoplast. Monolignol transport to the apoplast is not well understood. It may involve passive diffusion, although this may be targeted to sites of lignin initiation/polymerization by ordered complexes of both biosynthetic enzymes on the cytosolic side of the plasma membrane and structural anchoring of proteins for monolignol oxidation and polymerization on the apoplastic side. We present several hypothetical models to illustrate these ideas and stimulate further research. These are based primarily on studies in model systems, which may or may not reflect the major lignification process in forest trees.

## 1. Why the interest in lignin?

Lignin is the most abundant aromatic polymer produced in plants. It provides mechanical support and facilitates transport of water and solutes through the vascular system [1], as well as playing an important role in both passive and active defence [2–5]. By acting as a pre-existing physical barrier against pathogen invasion and herbivory, lignin also contributes to the recalcitrance of lignocellulosic biomass for biorefining [6].

Lignin polymers (figure 1*a*) generally consist of *p*-hydroxyphenyl (H), guaiacyl (G) and syringyl (S) units, derived from the three primary monolignols *p*-coumaryl, coniferyl and sinapyl alcohols, respectively (figure 1*a,b*) [1]. The biochemical reactions leading to these building blocks have been elucidated, although the pathway has often been revised and is likely to be nonlinear (figure 2*a*). Increased attention has been directed towards understanding lignin biosynthesis and its regulation over the past 10 years, largely driven by the aims of modifying the amount or composition of the polymer in plant cell walls to improve forage digestibility, facilitate bioprocessing of lignocellulose to liquid biofuels, or tailor the polymer itself for conversion to materials and/or bioproducts [9–15]. These studies have drawn attention to the major gaps in our understanding of the processes by which the precursors of lignin

**Figure 1.** Lignin and its components. (*a*) Schematic structure of a generic dicot lignin, showing some of the different linkage types between guaiacyl (G, derived from coniferyl alcohol) and syringyl (S, derived from sinapyl alcohol) units. The major β-*O*-4 linkages are shown in red. (*b*) The five hydroxycinnamyl alcohols derived from the monolignol pathway: 4-coumaryl alcohol (**H**), coniferyl alcohol (**G**), caffeyl alcohol (**C**), 5-hydroxyconiferyl alcohol (**5HG**) and sinapyl alcohol (**S**). (*c*) The structure of C-lignin, a caffeyl alcohol homopolymer found primarily in seed coats of certain non-crop species, with the units linked via benzodioxane structures (in red). (*d*) Recently discovered non-canonical components of lignin: **1**, tricin; **2**, resveratrol; **3**, coniferyl ferulate; **4**, tyramine ferulate.

are formed within the cell, transported to the intercellular space and polymerized with tissue- and cell-type specificity. Furthermore, it is unclear how models for the temporal and spatial control of lignification in some cell types can be extrapolated to other cell types or environmental conditions.

## 2. Lignin is not what it used to be

Deciphering the structures of complex phenolic-containing plant polymers such as lignin, suberin or sporopollenin presents many challenges [15–17]. In the case of lignin, which is arguably the least chemically heterogeneous of these three apoplastic polymers, the difficulties arise from variations in monomer unit composition and linkage types (figure 1*a*), chain length, degrees of branching (figure 1*a*) and developmentally or environmentally controlled differences in the above [15]. Over the past 15 years, two opposite schools of thought (with some taking the middle ground!) have either argued that lignin polymerization is a completely random process, driven only by the chemical propensities of the monolignols to couple such that the final lignin structure is dependent solely on their provision into the cell wall [18], or that lignin structure is finely controlled by some type of structural protein (template?)-driven mechanism [19]. Although there is much recent evidence in favour of the former hypothesis for synthesis

of bulk lignin, elements of the second model may help to explain the processes of lignin chain initiation.

Contrary to the traditional view, it is now clear that the H, G and S monolignols are not the only compounds that can be incorporated into lignin. Non-traditional monolignols fall into two categories: those that arise as a result of genetic modification of the monolignol pathway and those that occur naturally. There is, however, some overlap between these. The loss of function of caffeic acid/5-hydroxyconiferaldehyde 3/5-*O*-methyltransferase (COMT) (figure 2*a*) results in accumulation of high levels of 5-hydroxyguaiacyl (5HG; figure 1*b*) units in lignin [20], but 5HG lignin also occurs naturally in high proportion in the seed coats of at least one cactus species [21]. Lignin consisted almost entirely of the aldehyde derivatives of the common monolignols accumulates following loss of function of the major cinnamyl alcohol dehydrogenase (*CAD*) gene (figure 2*a*) in the model legume barrel medic (*Medicago truncatula*) [22]. C-lignin, a homopolymer of caffeyl alcohol (figure 1*c*) linked exclusively via benzodioxane units, is found in the seed coats of a number of species including vanilla orchid (*Vanilla planifolia*), *Jatropha* species and various cacti [21,23,24]. Although it has not proven possible to date to engineer a significantly C-unit-enriched lignin polymer in the vegetative tissues of plants, genetically engineered plants tolerate high levels of H, 5HG and H/G/S aldehydes [20,22,25,26] in their lignins,

royalsocietypublishing.org/journal/rsob    Open Biol. 9: 190215

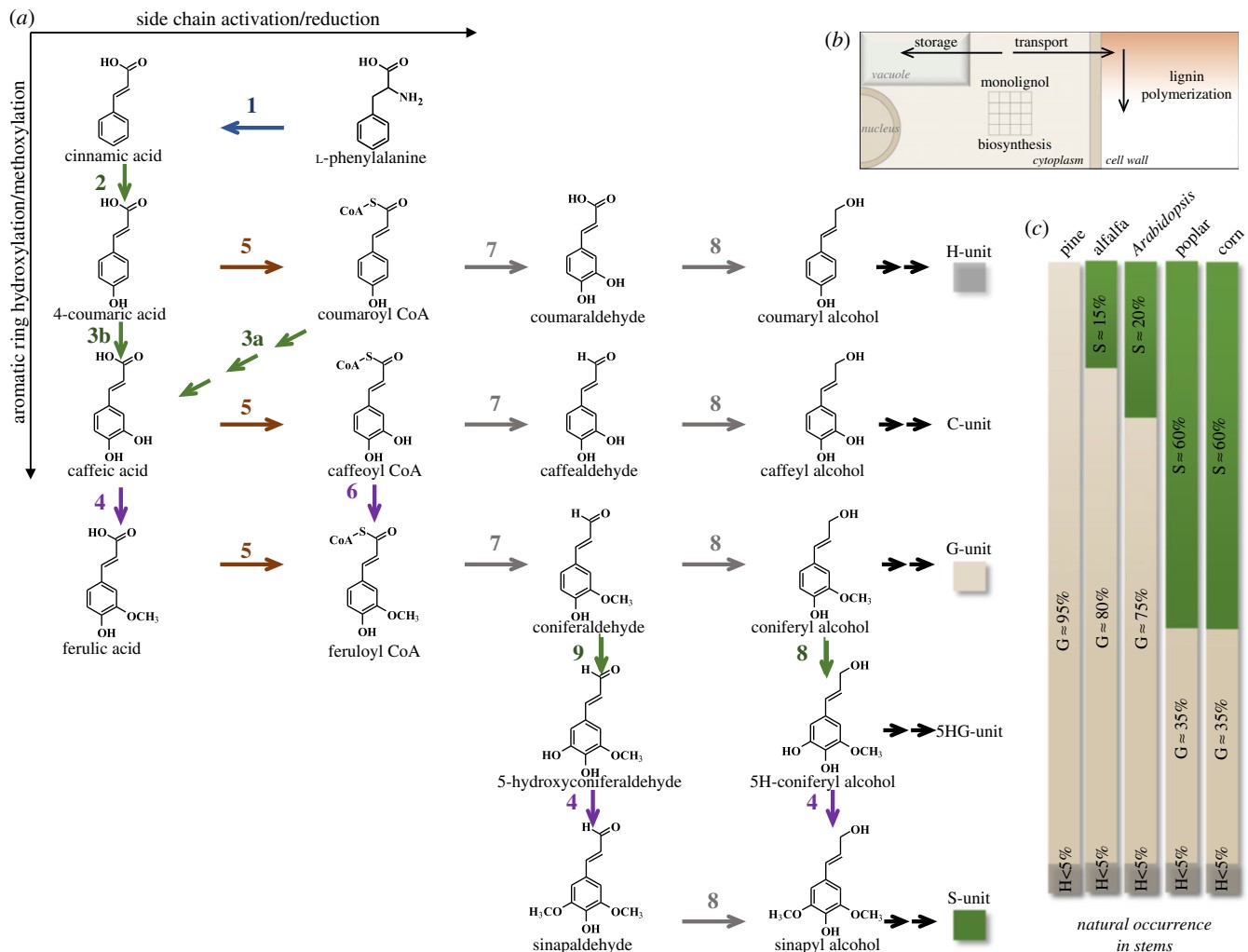

**Figure 2.** Pathways to monolignols and lignin diversity in plants. (*a*) A schematic depiction of the enzymatic reactions involved in monolignol biosynthesis from L-phenylalanine. The different types of reaction are colour-coded: blue, deamination; green, hydroxylation; purple, *O*-methylation; brown, CoA activation; grey, reduction. Enzymes are: 1, L-phenylalanine ammonia-lyase (PAL); 2, cinnamic acid 4-hydroxylase (C4H); 3a, the SS (figure 3) involving hydroxycinnamoyl CoA: shikimate hydroxycinnamoyl transferase (HCT), coumaroyl shikimate 3′-hydroxylase (C3′H) and CSE; 3b, coumarate 3-hydroxylase (C3H); 4, caffeic acid/5-hydroxy-coniferaldehyde 3/5-*O*-methyltransferase (COMT); 5, 4-hydroxycinnamate CoA ligase (4CL); 6, caffeoyl CoA 3-*O*-methyltransferase (CCoAOMT); 7, cinnamoyl CoA reductase (CCR); 8, cinnamyl alcohol dehydrogenase (CAD); 9, ferulic acid/coniferaldehyde 5-hydroxylase (F5H). (*b*) Cellular sites of monolignol synthesis and accumulation; monolignols are synthesized in the cytosol, with some enzymes exhibiting membrane attachment (figures 3 and 4), some monolignol derivatives accumulate in the central vacuole and the bulk of the monolignol pool is targeted to the apoplast for polymerization to lignin. (*c*) Variation in lignin monomer composition between species. Data compiled from [7,8].

perhaps contrary to expectations of a template mechanism for lignin biosynthesis [19].

The repertoire of molecules that can be incorporated into lignin continues to grow, although the non-classical components are generally not present in high amounts. Among the naturally occurring molecules, the flavone tricin (figure 1*d*) is of particular interest. This now appears to be present in all grass lignins, where it has been proposed to serve as a polymerization initiator unit [27]. The aromatic B-ring of tricin has the same substitution pattern as the S monolignol, resulting in 4-*O*-coupling with a monolignol at its β-position. The same *O*-methyltransferase enzyme appears to be responsible for addition of the second *O*-methyl group in the biosynthesis of both tricin and the S-lignin precursor sinapaldehyde [28]. Hydroxystilbenes and/or their glycosides, including the much studied nutraceutical compound resveratrol (figure 1*d*), are found naturally in the lignin from palm (including coconut, *Cocos nucifera*) fruit endocarp and Norway spruce (*Picea abies*) [29,30], and the nitrogen-containing compounds tyramine ferulate (figure 1*d*) and diferuloylputrescine are

present in the lignin of transgenic tobacco (*Nicotiana tabacum*) plants downregulated in cinnamoyl CoA reductase (CCR) (figure 2*a*) and in corn (*Zea mays*) pericarp, respectively [31,32].

Monolignol ferulate and coumarate conjugates (figure 1*d*) are now also viewed as natural components of lignins; the latter are particularly rich in monocot lignins [15]. Their presence inspired the design of a synthetic biology strategy to introduce ester bonds into lignin to render the molecule more amenable to chemical cleavage [33]. A feruloyl CoA monolignol transferase (*FMT*) gene from dong quai (*Angelica sinensis*) that catalysed the formation of the ferulate esters of coniferyl and sinapyl alcohols was expressed in the xylem of transgenic hybrid poplar (*Populus alba* x *grandidentata*), resulting in lignin (so-called zip-lignin) with intramolecular ester linkages which are more easy to cleave chemically than the usual inter-unit linkages [33]. In addition to its biotechnological utility, this work showed that, as with classical monolignols, dimeric lignin precursors could be formed in the cytosol, transported to the apoplast (without instead being sequestered in the vacuole) and incorporated into lignin (figure 2*b*) without the need for

royalsocietypublishing.org/journal/rsob   Open Biol. 9: 190215

additional genes beyond the added acyltransferase. No other plant polymer appears to exhibit such structural flexibility, suggesting that, if natural and synthetic lignin building blocks have certain chemical features, both transport to the apoplast and subsequent polymerization are generally facile processes. Additional flexibility is suggested by the widely divergent lignin monomer compositions of different plant species, with gymnosperms such as pine (*Pinus* spp.) possessing lignin made of only G units, monocot species such as *Z. mays* having a preponderance of S units, and widely varying S/G ratios in dicotyledonous plants such as alfalfa (*Medicago sativa*), thale cress (*Arabidopsis thaliana*) and poplar (*Populus* spp.) (figure 2*c*).

# 3. Yet more revisions to the monolignol pathway

## 3.1. The enigma of the esters pathway to monolignols

The biosynthetic pathway to the major monolignol precursors of lignin has undergone several revisions since it was first proposed in the 1960s and 1970s [34–36]. Generation of the classical H, G and S monolignols involves sequential hydroxylation and methoxylation of the aromatic ring coupled with side-chain reduction of the acid to the alcohol (figure 2). The enzymes required comprise a deaminase (L-phenylalanine ammonia-lyase, PAL) to first convert L-phenylalanine to cinnamic acid, followed by a series of hydroxylases, *O*-methyltransferases and reductases. After the demonstration of all these enzyme activities and the cloning of their corresponding genes between the early 1960s and around 2001, the main debate centred on the 'ordering' of the reactions, with arguments as to whether they formed a metabolic grid or were conscribed to a more linear arrangement. An arrangement of both linear and grid-like sections to the pathway was generally accepted after the discovery that the 5-hydroxylation of ferulic acid *in vitro* was kinetically unfavoured, with the 5-hydroxylation preferentially occurring at the level of coniferaldehyde, and that the *O*-methylation of the 5-hydroxyferuloyl moiety *in vitro* occurred most efficiently at the level of the aldehyde or alcohol, both these reactions being catalysed by the enzyme that had first been identified as caffeic acid 3-*O*-methyltransferase (COMT) [35] (figure 2). Both biochemical and genetic studies then indicated that the 3-hydroxylation of the coumarate moiety occurred at the level of a shikimate ester, formed by trans-esterification of coumaroyl CoA (also a precursor of flavonoid compounds) by the enzyme hydroxycinnamoyl CoA: shikimate hydroxycinnamoyl transferase (HCT) [36] (figure 3). The aromatic ring on this shikimate ester was shown to be hydroxylated by the cytochrome P450 CYP98A3 (coumaroyl shikimate 3′-hydroxylase, C3′H) [37,38] (figure 3). Mutation of the corresponding gene (the so-called reduced epidermal fluorescence (*REF*) *8* gene in *A. thaliana*) led to stunted plants with strongly reduced lignin levels comprised an unusually high percentage of H units [39,40]. The caffeoyl shikimate product of C3′H was proposed to be converted back to caffeoyl CoA by HCT operating in the reverse direction, thus completing a series of reactions termed the 'shikimate shunt' (SS) or the 'esters pathway' (figure 3). The C3′H and HCT enzymes of the SS are present in dicot and monocot species, and also in gymnosperms [41].

The 'reverse' HCT reaction to form caffeoyl CoA is kinetically much less favourable than the forward reaction to form coumaroyl shikimate [42], and its operation *in vivo* has also been questioned recently based on mathematical modelling [43]. This problem appeared to be solved by the discovery of a caffeoyl shikimate esterase (CSE) that hydrolyses caffeoyl shikimate (and to a lesser extent coumaroyl shikimate) to the free acid [36] (figure 3). That this enzyme is important for monolignol biosynthesis is clear from genetic analysis in the model dicot plants *A. thaliana* and *M. truncatula*, where the loss of function of CSE results in dwarf plants with reduced levels of G and S units in their lignin [36,44]. RNAi-mediated downregulation of the two *CSE* genes expressed in poplar xylem also results in reduced lignin amount, accompanied by improved biomass saccharification efficiency [45]. CSE therefore appears to function as the final enzyme in the SS, with caffeic acid as the product (figure 3). However, *CSE* genes are absent in some grasses such as purple false brome (*Brachypodium distachyon*) and rice (*Oryza sativa*) [44]. CSE is therefore the only example of a monolignol pathway enzyme that is dispensable for lignin biosynthesis. In its absence, the reverse HCT reaction would, in spite of its unfavourable kinetics, appear to be necessary to generate caffeoyl CoA (figure 3).

The involvement of CSE in the biosynthesis of caffeoyl CoA requires, in comparison to the SS with the reverse HCT reaction, the consumption of an extra molecule of ATP (figure 3). So why do plants use this complex pathway to achieve a simple hydroxylation reaction? One hypothesis proposes that phenylpropanoid pathway intermediates are 'tagged' by conjugation to divert them into specific areas of metabolism; for example, hydroxycinnamate esters targeted to the cell wall [46]. The existence of a wide range of esters of phenylpropanoids with CoA, shikimate, quinate and glucose, to name a few, is not inconsistent with this hypothesis, but the hypothesis lacks genetic support. It is more likely that the involvement of shikimate esters in lignin biosynthesis is linked to the fact that shikimate is the key precursor for the biosynthesis of aromatic amino acids in the chloroplast (figure 3). An estimated 30% of all fixed carbon is directed through the shikimate pathway to proteins and a range of specialized metabolites [47]. Because of the large amount of L-phenylalanine that is converted to lignin during vascular differentiation, the concentration of the cytosolic shikimate pool could act as a sensor of the flux into lignin. This has been proposed to occur via a simple 'precursor shut-off valve' whereby the SS attenuates flux into the monolignol pathway when shikimate, and by extension phenylalanine, concentrations are low, thereby reserving phenylalanine for protein biosynthesis [43], or perhaps production of other phenylpropanoids such as flavonoids [48], which are derived from coumaroyl CoA. The major piece missing from this puzzle is the mechanism whereby shikimate exits the chloroplast. To the best of the authors' knowledge, no transporters for shikimate efflux or influx into plastids have been described. A plastidial shikimate efflux transporter would appear to be critical for monolignol biosynthesis, so it is perhaps surprising that a gene encoding such a transporter has not been identified in mutant screens. Recent studies have shown that plants possess a cytosolic route to L-phenylalanine [49], although this does not currently appear to involve steps pre-chorismate. Determining whether shikimate is synthesized in or transported into the cytosol will be important for completing our understanding of the operation and function of the SS.

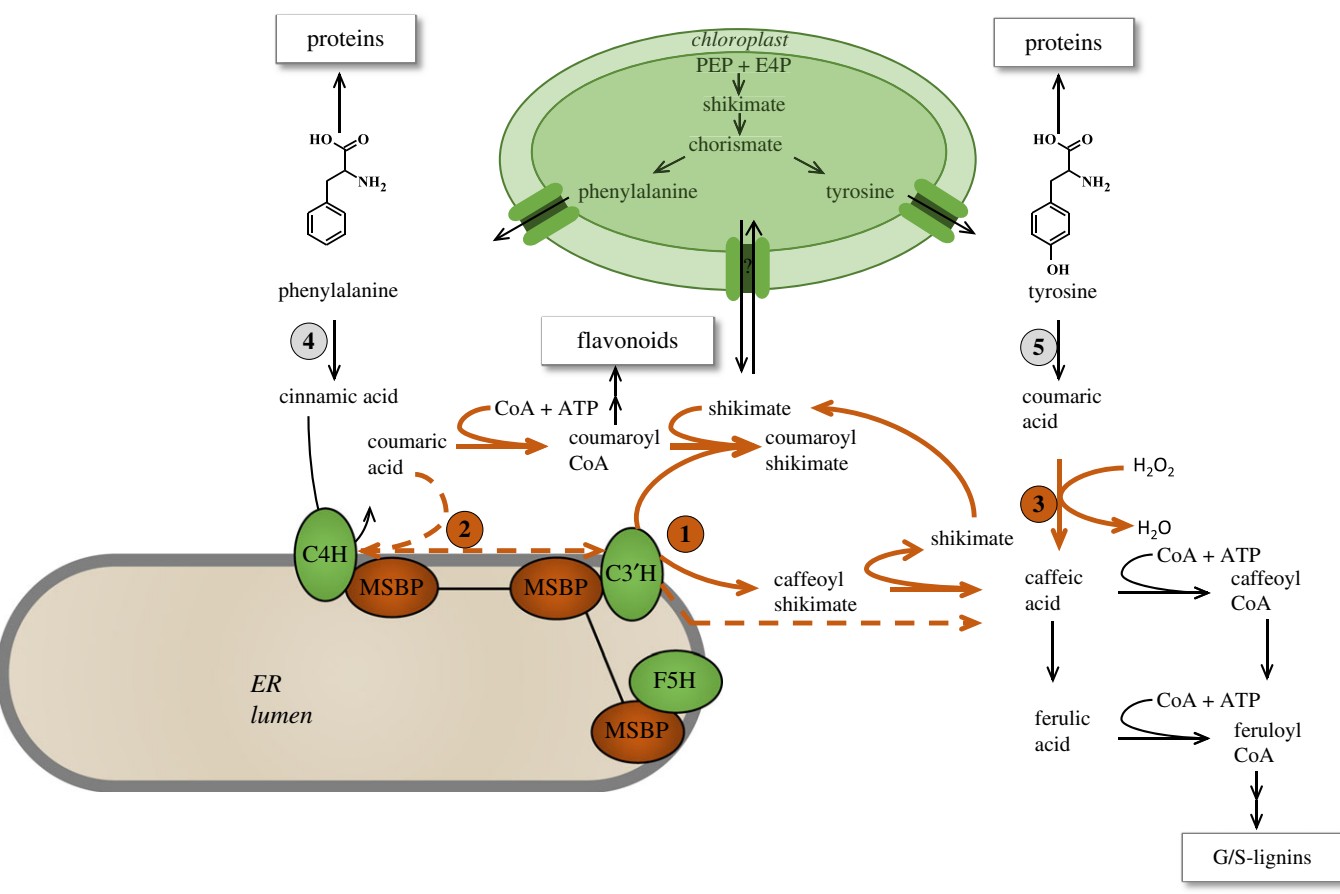

**Figure 3.** Model showing the central role of the hydroxylation of coumaric acid in the pre-G portion of the monolignol pathway. Reactions shown with orange arrows are associated with conversion of coumaric acid to caffeic acid or caffeoyl CoA. The three pathways for coumarate hydroxylation are: **1**, by C3′H via the SS involving the actions of HCT, C3′H and CSE; **2**, by a complex of C4H and C3′H (dashed line); **3**, by the soluble cytosolic ascorbate peroxidase C3H/APX. Coumarate is formed from phenylalanine by the sequential activities of PAL (**4**) and C4H, or, in grasses, also directly from L-tyrosine by the action of TAL (**5**). Unless otherwise indicated, reactions occur in the cytosol. Shikimate, formed from phosphoenol pyruvate (PEP) and erythrose 4-phosphate (E4P), is an intermediate in the formation of L-phenylalanine and L-tyrosine in the chloroplast, but also acts as acyl donor for the trans-esterification reactions of the SS for coumarate hydroxylation in the cytosol. The cytochrome P450 enzymes catalysing hydroxylation of cinnamic acid (C4H) and coumaroyl shikimate (C3′H) are anchored to the cytosolic face of the ER. F5H is the cytochrome P450 that catalyses the hydroxylation of ferulate, 5-hydroxyconiferaldehyde and 5-hydroxyconiferyl alcohol in the biosynthesis of S-lignin (figure 2). MSBP, membrane sterol binding protein(s) that form linkers between the three cytochrome P450s of the monolignol pathway.

Small non-coding RNAs play a role in the control of lignification. For example, laccases involved in lignin polymerization are under the control of micro-RNAs [50]. The roles of transfer RNA-derived fragments (tRFs), also known as ribosome-associated non-coding RNAs (ranc RNAs), as another class of biological regulators have been actively investigated in microorganisms and animal cells [51,52]. Ranc RNAs also occur in plants [53]. In view of the importance of the nexus between deployment of L-phenylalanine for the synthesis of protein (through phenylalanyl tRNA) and lignin (through cinnamic acid), it will be interesting to determine whether translational regulation by tRFs RNAs is another component in the regulation of this critical metabolic bifurcation.

## 3.2. Old routes to lignin revisited

Ever since the discovery of PAL in the 1960s, it was known that some PAL enzymes from grasses also possessed L-tyrosine ammonia-lyase (TAL) activity, although the physiological significance of this was unclear. *Brachypodium distachyon* possesses 8 *PAL* genes, one of which encodes a bi-functional PAL/TAL (PTAL) [54]. Genetic analysis recently has shown that this enzyme is preferentially, but not exclusively, involved in the biosynthesis of S-lignin, and isotopic labelling

studies have revealed that approximately 50% of the lignin in *B. distachyon* stems is synthesized from L-tyrosine rather than from L-phenylalanine. Isotope dilution experiments have suggested that the pools of coumaric acid derived directly as a product of soluble TAL activity or via the consecutive activities of PAL and endomembrane-bound C4H (figure 3) may not be fully equivalent [54]. The consequences of this require further study.

In addition to the possession of PTAL, there is another important difference between monocots and dicots as regards the early steps in monolignol biosynthesis. Downregulation of the SS pathway enzymes HCT or C3′H by RNA interference has far less effect on lignin content/composition in some monocots such as *B. distachyon* [55] and switchgrass (*Panicum virgatum*) [56,57] than in dicots [37,40,58]. This implies the existence of a pathway that can bypass the SS. The simplest model invokes a second hydroxylase (C3H) that carries out the 3-hydroxylation reaction on free coumaric acid (figure 3). Early studies showed the presence of such a hydroxylase activity in crude extracts from several plant species [59–61], but corresponding genes were not subsequently identified. Instead, it was proposed that the conversion of coumarate to caffeate is facilitated by a trimeric complex of two individual CYP73A (cinnamic acid 4-hydroxylase, C4H) protein

molecules and one CYP98A (C3′H) protein molecule in poplar xylem [62] (figure 3). Association between C3′H and C4H was reported to massively increase the turnover of the former enzyme, and also allowed for direct hydroxylation of 4-coumarate. Further evidence for associations between these enzymes on the cytosolic face of the endoplasmic reticulum (ER) at high degrees of oligomerization, with weaker association of HCT with the complexes, was provided by fluorescence imaging microscopy in leaves of both *A. thaliana* and *Nicotiana benthamiana* [63], but these studies did not address biochemical activities of the complexes. The concept of dynamic complexes representing metabolons for channelling of pathway intermediates has been a central, but generally unresolved, concept in plant specialized metabolism [64–66]. In the case of C4H/C3′H, genetic analysis of the *in vivo* catalytic capabilities of the proposed complex(es) appears difficult because the amino acid residues that might facilitate oligomerization are not known. Furthermore, it has recently been shown that the C4H/C3′H metabolon also involves the final cytochrome P450 enzyme of the monolignol pathway, ferulate 5-hydroxylase (F5H), through linking of the proteins by two ER membrane-resident membrane-steroid binding proteins (MSBPs) [67] (figure 3). These linker proteins are regulated by the same transcription factor that controls the expression of F5H [67], and loss of function of both MSBPs results in a post-translationally controlled reduction in the P450 enzyme activities. Importantly, the assembly of the enzyme complex via MSBPs does not involve direct interactions between the P450 s [67], so it is not clear how this type of complex alters the biochemical functions of the composite enzymes, as proposed for C4H/C3′H in poplar (*Populus trichocarpa*) [62].

A path to caffeate that bypasses membrane-associated complexes was recently identified through a re-evaluation of the early reports of a soluble coumarate hydroxylase. The activity in corn roots was shown to belong to a bi-functional cytosolic ascorbate peroxidase/coumarate 3-hydroxylase (APX/C3H) [68], an enzyme that had been extensively studied (as APX1) in *A. thaliana* in relation to protection against reactive oxygen species (ROS) [69]. Genetic analysis in *A. thaliana* and *B. distachyon* demonstrated that partial loss of function of this C3H resulted in a reduction in lignin levels, with a stronger phenotype in the *cse* null mutant background in *Arabidopsis* [68].

The involvement of a soluble C3H essentially bypasses the HCT, C3′H and CSE reactions of the SS, and provides a pathway which, if initiated from L-tyrosine, also avoids C4H and therefore should have no association with the endomembrane system (figure 3). Kinetic competition studies indicated that the caffeic acid generated by C3H would be preferentially methylated to ferulic acid by COMT rather than converted to caffeoyl CoA (figures 2 and 3) [68], suggesting that the once favoured 'acids pathway' to monolignols should be re-instated, at least for the synthesis of G lignin.

The existence of parallel pathways in the pre-G portion of the monolignol pathway potentially provides flexibility to phenylpropanoid biosynthesis under different environmental conditions. Synthesis of flavonoids is strongly induced by both biotic and abiotic stress [70], and this synthesis would compete with lignification for coumaroyl CoA during development under stress (e.g. high light) conditions. Direct conversion of coumarate to caffeate would bypass the diversion of flux to flavonoids and related compounds (figure 3). It could also provide a more energetically efficient route to caffeoyl CoA in plants which possess CSE activity. The effective operation of such a model presupposes some kind of cross-talk between the soluble and membrane-associated pathways, the nature of which remains to be determined. To this end, it will be informative to evaluate metabolic flux through the different branches of the monolignol pathway under controlled application of abiotic stress.

# 4. Monolignol biosynthesis and reactive oxygen

The involvement of an ascorbate peroxidase in monolignol biosynthesis suggests a potential link between formation of monolignol substrate in the cytosol and detoxification of hydrogen peroxide, the agent that is also responsible for monolignol radical formation catalysed by peroxidases in the cell wall. Ascorbate peroxidases are present in different cellular compartments in plants, with a major function of detoxifying hydrogen peroxide generated during abiotic stress [71]. C3H is the third enzyme in the monolignol pathway with an additional function related to ROS, the others being CSE and CCR (see below). It seems unlikely that this is coincidental.

Based on our current understanding of the evolution of the monolignol pathway, PAL was acquired by the earliest land plants by horizontal gene transfer from soil bacteria [72], and the initial function of phenylpropanoid compounds was in protection against microbial attack and abiotic stress [73]. The need for protection of the first land plants from oxidative damage induced by high light, desiccation, etc., probably arose prior to the need for a structural polymer providing the physical strength for plants to grow towards the light. It is possible that the alternative reactions of some monolignol pathway enzymes reflect their original functions in stress protection, and that this multifunctionality has been maintained to allow for effective regulation of the supply and utilization of ROS for lignin biosynthesis.

Before the role of CSE in monolignol biosynthesis was established, the protein was recognized as a lysophospholipase involved in the turnover of lysophospatidylcholine, formed by oxidative damage of membrane lipids [74]. The *A. thaliana* lysophospholipase 2 (CSE/LPL2) interacts with acyl CoA-binding protein 2 (ACBP2) through the ankyrin repeats on the latter protein, resulting in enhanced binding of lyosphospatidylcholine and localization of the complex at the plasma membrane where it functions in membrane repair (figure 4) [74,75]. The expression of lysophospholipase 2/CSE is induced by hydrogen peroxide (figure 4), and its loss of function and over-expression result in decreased and increased sensitivity to hydrogen peroxide, respectively [74]. To the best of our knowledge, these properties of this enzyme complex have not been considered in relation to its involvement in monolignol biosynthesis. For example, although members of the plant ACBP2 family exhibit relatively broad substrate binding specificity [76], it is not known whether they can bind hydroxycinnamoyl CoAs. The induction of lysophospholipase 2/CSE by hydrogen peroxide could conceivably help balance monolignol synthesis with hydrogen peroxide availability, and targeting of CSE to the plasma membrane through interaction with ACBP2 is of particular interest in view of the potential localization of subsequent steps in the pathway, and their association with generation of hydrogen peroxide as described below.

royalsocietypublishing.org/journal/rsob    Open Biol. 9: 190215

royalsocietypublishing.org/journal/rsob  *Open Biol.* **9**: 190215

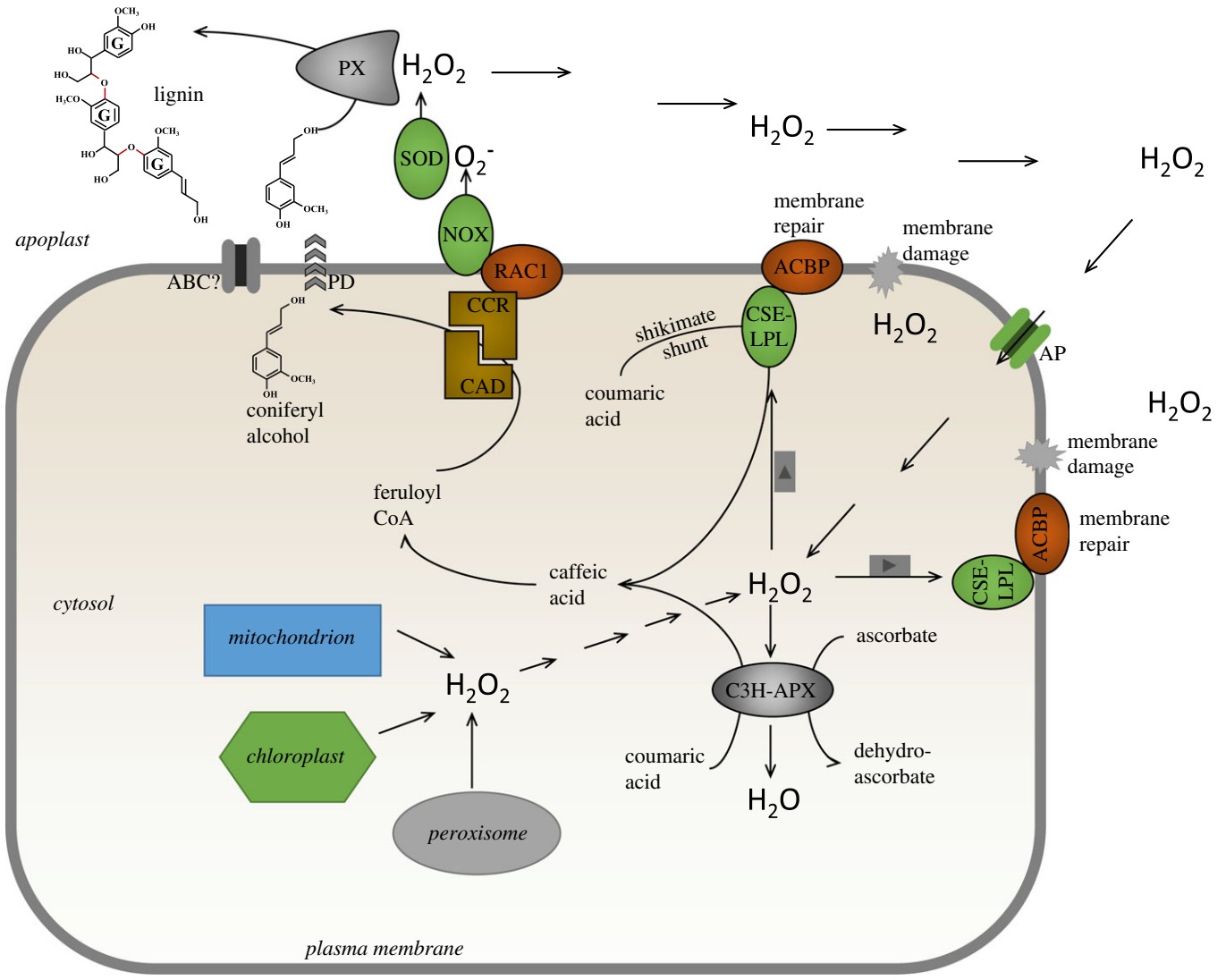

**Figure 4.** A model linking monolignol (coniferyl alcohol only for simplification) biosynthesis to generation and detoxification of hydrogen peroxide. Hydrogen peroxide is generated in multiple intracellular compartments (including chloroplast, mitochondrion and peroxisome), with a major source being the plasma membrane-associated NADPH oxidase (NOX) with its catalytic site in the apoplast. NOX is activated by binding of the monolignol pathway enzyme cinnamoyl CoA reductase (CCR) to the plasma membrane-associated RAC1 GTPase, and CCR and cinnamyl alcohol dehydrogenase (CAD) exist in a complex that may therefore target monolignol (shown here as coniferyl alcohol) to the apoplast via 'directed' passive diffusion (PD) or possible via membrane transporters (here depicted as an ABC family transporter). Apoplastic hydrogen peroxide derived from superoxide via superoxide dismutase (SOD) is used for monolignol oxidation via peroxidases. Hydrogen peroxide and other ROS cause membrane damage by oxidation of lipids, and this damage can be repaired by lysophospholipase 2, which is also the caffeoyl shikimate esterase (CSE/LPL2) of monolignol biosynthesis via the SS. CSE/LPL2 is attached to the plasma membrane via the ankyrin repeats of the acyl CoA-binding protein (ACBP), and this localization may further assist in moving the biosynthesis of monolignols to the inner side of the plasma membrane for directed diffusion to the apoplast. Hydrogen peroxide generated in the cytosol or transported from the apoplast via aquaporins (AqP) can be detoxified by cytosolic ascorbate peroxidase, with conversion of ascorbate to monodehydroascorbate. This enzyme also functions to convert coumaric acid to caffeic acid (C3H/APX), thereby bypassing the SS. A concentration gradient across the plasma membrane would appear to be necessary to allow apoplastically generated hydrogen peroxide to act as a signal for coordinating monolignol generation (in the cytosol) and polymerization (in the apoplast).

The final steps in monolignol biosynthesis that generate the side-chain alcohol function are sequential reductions in an acyl CoA (for G units, feruloyl CoA) catalysed by CCR and CAD (figures 2 and 4). *In vitro* studies suggest kinetic coupling between these two enzymes [77]. Mathematical modelling in multiple species [78–80], and direct protein interaction studies with enzymes from developing poplar xylem [81], have suggested that CCR and CAD may operate in a membrane-associated complex, the formation and dissociation of which may be dynamic. In computational models, the membrane compartment was presented as the outer surface of the ER [80], but could equally be the inner surface of the plasma membrane. Indeed, earlier studies had indicated that OsCCR2 from *O. sativa* can localize to the plasma membrane through

interaction with OsRAC1, a member of the Rac/Rop family of small membrane-bound GTPases that is a positive regulator of hydrogen peroxide production via the NADPH oxidase (NOX) belonging to the class of the respiratory burst oxidase homologue (RBOH) proteins [82] (figure 4). The interaction between OsRAC1 and OsCCR2 resulted in an approximately 10-fold increase in the turnover of feruloyl CoA to coniferaldehyde by OsCCR2; furthermore, OsCCR2 bound to OsRAC1 only in the presence of GTP, which was also necessary for the stimulation of NOX by OsRAC1 [82]. Activation of NOX generates the hydrogen peroxide required for monolignol oxidation and polymerization via peroxidases (PX, figure 4). It should be cautioned that the induction of OsCCR2 and OsRAC1 is primarily associated with lignification during

royalsocietypublishing.org/journal/rsob    Open Biol. 9: 190215

plant defence, and many plants have at least two *CCR* genes, with both specialized and redundant functions. The loss of function of *A. thaliana CCR1*, implicated in developmentally regulated lignification, results in a marked reduction in ROS [83]. The authors interpreted this reduction to result from increased levels of the antioxidant ferulic acid, but the data are also consistent with AtCCR1 as an agonist for NOX activity, with subsequent hydrogen peroxide generation.

APX1/C3H uses cytosolic pools of hydrogen peroxide, which would also be necessary to induce lysophospholipase 2/CSE, unless this effect is indirect. Production of hydrogen peroxide for monolignol polymerization, via superoxide generated by NOX, occurs in the apoplast (figure 4). Any model that links monolignol generation with monolignol polymerization through ROS must include a mechanism for transport of hydrogen peroxide or its precursor superoxide across the plasma membrane from apoplast to cytosol. It is unlikely that superoxide is sufficiently protonated at the pH of the plant apoplast to diffuse efficiently across the plasma membrane [84]. Of the many classes of plant superoxide dismutases (SOD) that generate hydrogen peroxide from superoxide, some Cu/Zn SODs are localized to the apoplast [85,86], and have been proposed to be involved in generation of hydrogen peroxide for lignification. Diffusion of hydrogen peroxide across membranes is slow, and transport is believed to occur primarily via water channels (aquaporins, AqP) (figure 4) [87,88], some of which are weakly inducible by hydrogen peroxide in *A. thaliana* [89]. Testing whether active oxygen species serve as a link between monolignol biosynthesis in the cytosol and polymerization in the apoplast will be difficult because of the effects of highly toxic superoxide build-up in the cell walls of knock-out lines of superoxide dismutase [90], the short half-life of hydrogen peroxide in biological systems [91] and the presence of alternative sources of ROS from organelles in the cytosol (figure 4) [92]. Furthermore, inward diffusion of hydrogen peroxide from the apoplast will require a concentration gradient across the plasma membrane. This may occur following activation of NOX (RBOHs) during plant defence, but it is less clear if such gradients exist during developmental lignification.

# 5. Monolignol transporters: cases for and against

It is surprising that, in spite of evidence suggesting ATP- and protein-dependent biochemical processes, there is still no consensus as to the mechanism of monolignol transport across the plasma membrane [93]. By contrast, although transport of other phenolic compounds such as flavonoids may occur by complex mechanisms, the membrane transporters involved have proven to be genetically tractable (e.g. [94]). Based on the widespread occurrence of monolignol glycosides in some plant species, and the well-known requirement for glycosylation to facilitate transport of plant polyphenols into the cell vacuole, it has been suggested that monolignol glycosides may also be transport forms for movement into the cell wall [95]. However, the evidence that this is an active process in living tissues is far from conclusive, and such glycosides might, like quinate esters, function more as vacuolar storage forms [96]. Indeed, it is clear that a range of glycosides of monolignols and their oligomers are localized in the central vacuole in *Arabidopsis* leaves [97].

Mechanisms for monolignol transport currently under consideration range from involvement of specific transporters to trafficking through vesicles or simple passive diffusion, either directly through the plasma membrane or through specific pores [93]. Perhaps different mechanisms occur in different cell types or under different physiological conditions. Surprisingly, there is only a single report of a specific transporter for any monolignol species, an ABC transporter from *A. thaliana* that transports coumaryl alcohol, but that does not transport the more abundant coniferyl and sinapyl alcohols [98]. We have recently proposed that the transport of monolignols for polymerization in the apoplastic space is likely to occur by passive diffusion across the plasma membrane [99]. Molecular dynamic simulations supported passive membrane permeation of 69 different monolignols, dimeric lignin-related phenolics and related compounds, including tricin, across a model membrane representative of *Z. mays* [99]. Charged or glycosylated molecules do not diffuse in this way. The calculated rates of transport are sufficient to support lignin biosynthesis, particularly if these are increased by oligomerization of the mono-/di-lignols on the apoplastic side of the membrane. Transport of artificial lignin monomers, including dimers such as coniferyl ferulate, is accommodated by this model. The main issue is how to explain the monodirectional diffusion of monolignols across only the plasma membrane. The simplest explanation invokes an asymmetric concentration distribution across the plasma membrane as a result of oligomerization of the monolignols by localized complexes within the apoplastic space [100]. Although tricin can theoretically be transported by passive diffusion, flavonoids are usually transported across membranes (i.e. the tonoplast for vacuolar storage) via vesicle trafficking, often as glycosides and commonly using energy-dependent MATE family transporters [101]. As a wide range of tricin glycosides are found in plants [102], it is not clear how tricin's directional transport to the cell wall is controlled.

It has been proposed that monolignols are exported out of the cell into all areas of the apoplast where they are highly mobile [103]. However, there may be different mechanisms of monolignol transport in living tissues such as the Casparian strip (CS) versus terminally dying cell types (e.g. xylem elements) during development, and in plant cells responding defensively to pathogen attack (e.g. programmed cell death and cooperative lignification in living cells (reviewed in [104]). We deliberately avoid making these distinctions in this article, and present more generic models. We make no claims as to whether these models are correct in all details but hope that they will stimulate further experimentation.

# 6. Lignin initiation versus polymerization

The oxidative polymerization of monolignols is catalysed by laccases (using molecular oxygen) and peroxidases (using hydrogen peroxide). The possession of large gene families encoding both groups of enzymes in plants has generally hindered the genetic dissection of monolignol polymerization; *A. thaliana* possess 17 laccase genes [105] and 73 class III peroxidase genes [106]. Recent findings have provided some clarification as to the general roles of these genes in monolignol polymerization, and allow a distinction to be made between processes of initiation and bulk polymerization. Analysis of the *lac4 lac11 lac 17* triple mutant of *A. thaliana*, which expresses a normal complement of peroxidases, but makes virtually no

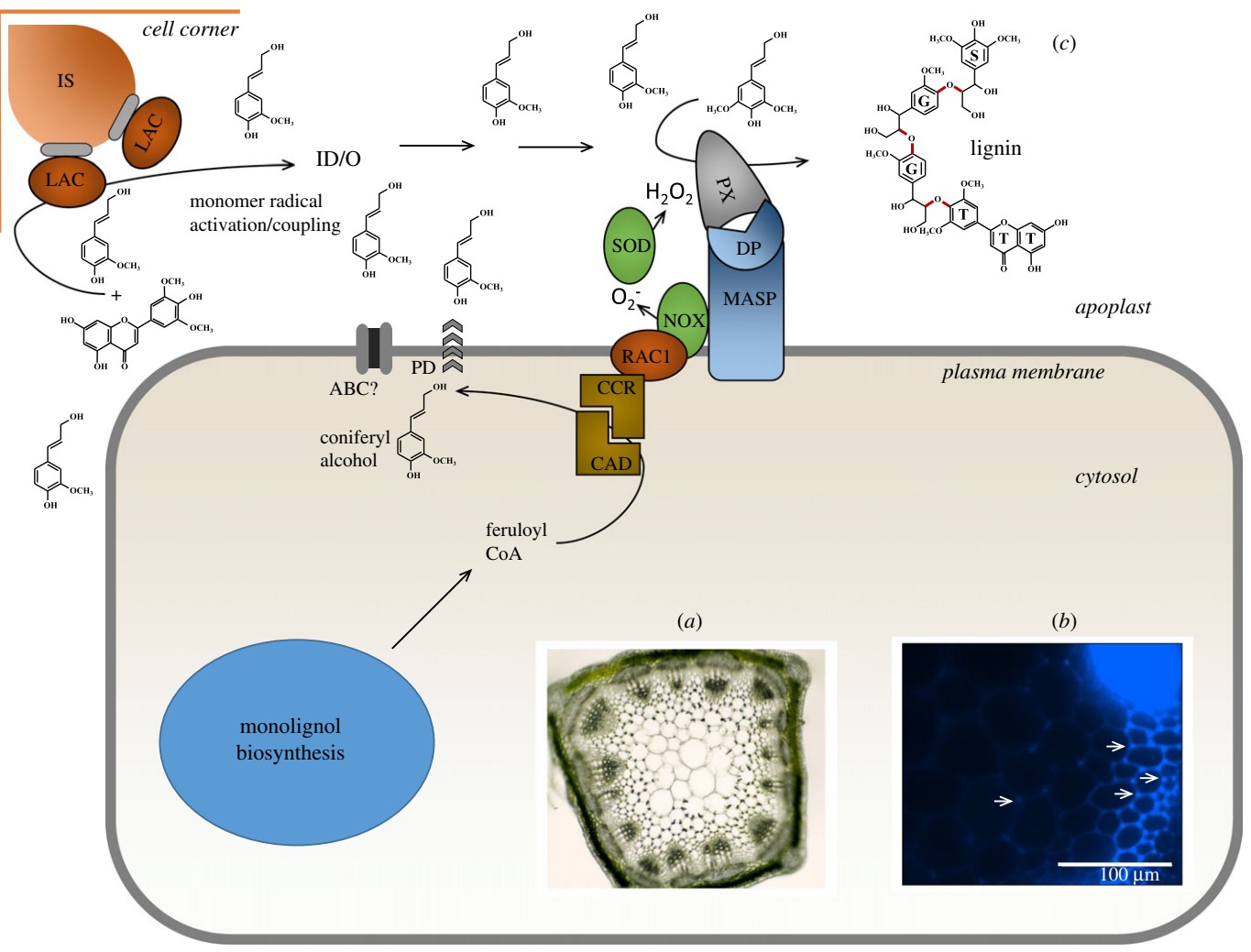

**Figure 5.** Initiation of lignification. (*a*) Cross-section through the stem of an alfalfa plant showing the onset of ectopic lignification in the pith due to antisense-mediated suppression of a WRKY family transcription factor [109]. Note spots of lignification in the corners of pith cells (arrows) that will eventually become fully lignified. (*b*) Magnified section of a stem cross-section for the same plant as shown in (*b*), viewed under UV light. Note blue lignin autofluorescence in the corners of the pith cells (arrows). Bar, 100 μm. Photographs are unpublished data of L. Gallego-Giraldo and R.A.D. (*c*) A generalized model for lignin initiation and polymerization within the apoplast (with coniferyl alcohol only for simplification). This composite model is designed to accommodate the genetic evidence implicating laccase in initiation of lignification, the finding of scaffold-mediated localization of peroxidases in the CS along with dirigent proteins (DPs), and the apparent localization of initial lignin accumulation to cell corners during both normal development and ectopic lignification in most plants, including trees. It is generic and does not reflect one specific cell type. Moreover, it does not accommodate possibilities of additional sources of apoplastic $H_2O_2$ [110]. Monolignols and potential initiator molecules enter the apoplast by passive diffusion, which may however be directed to sites of lignin polymerization through association of the terminal steps in monolignol biosynthesis with a membrane-bound activator of NOX (RAC1, as shown in figure 4). Monolignols are then assumed to diffuse freely within the apoplast. The initiation of lignification is depicted as laccase-mediated oxidative coupling of an initiator unit (shown here as tricin, but it could be a classical monolignol) with one or more monolignols to give an initiator dimer/oligomer (ID/O). The laccase is attached to an 'initiation site' (IS, probably a carbohydrate polymer) in the developing secondary cell wall, possibly through it glycan substituent. The ID/O is depicted as diffusing to the site where peroxidase catalysed polymerization of bulk lignin occurs. Peroxidase may be located near NOX for generation of hydrogen peroxide through association with MASPs, including Casparian strip domain-like proteins (CASPLs) and DPs. For other abbreviations, see caption to figure 4.

lignin in vascular tissues and fibres, although it has a normally lignified CS (see below), suggests that laccase may be essential for initiating lignification in vascular tissues, and that peroxidase alone cannot do the job [107]. Laccases are glycoproteins, on which the glycan moiety can account for around 30% of the molecular mass. Studies with both plant and fungal laccases have shown that removal of the glycan portion can have significant to no effect on the turnover rate and stability of the enzyme. Studies with an inducible tracheary element differentiation system in *A. thaliana* have suggested that monolignols are highly mobile within the apoplast, but that laccases are targeted to specific domains for initiation of lignification [103,108]. Lignin appears to be first deposited in cell corners, as seen clearly in interfascicular fibres of *A. thaliana* [105] and

ectopically lignifying pith cells of *M. sativa* (figure 5*a,b*). Laccase 4 and peroxidase 64 have been linked to lignification in *A. thaliana*, and Lac 4 was localized exclusively to the thick secondary cell wall, where it appeared to be immobile, whereas peroxidase 64 was localized mainly to the cell corners [108]. Studies are needed to determine whether the glycan is necessary for targeting laccase to a particular region within the cell wall, and, if so, whether this is determined during or after the process through which laccases are secreted, along with pectins and hemicelluloses, via the Golgi system.

The CS is a layer of lignified cells in the root endodermis that acts as a diffusional barrier. Here, lignification is dependent upon peroxidases but not laccases. Microscopic studies using chemical inhibitors in a series of mutant *A. thaliana*

lines revealed NOX in proximity with specific peroxidases through the involvement of plasma membrane-associated scaffold proteins (MASPs) also termed CS domain proteins (CASPs) [111] (figure 5c). This association was proposed to target the ROS species required for lignification (hydrogen peroxide derived from the dismutation of superoxide generated by NOX) to the region where monolignol substrate would be available [111]. Thus, the CASPs define the origin of lignin deposition in the CS [112]. A similar situation has recently been demonstrated in pathogen-induced lignification in *A. thaliana* leaves, where lignification is also critically dependent on the expression of CASPs [113]. It is tempting to speculate that the provision of monolignol substrate could also be spatially controlled by targeting the CCR/CAD complex to the cytosolic side of the plasma membrane via the system that activates NOX (the Rac/Rop GTPase) to further increase the efficiency of monolignol polymerization by avoiding transport of monolignols across internal membranes, minimizing oxidative stress [111] and potentially coordinating monolignol biosynthesis and hydrogen peroxide availability (figures 4 and 5c). Whether this model could also operate during developmentally regulated lignification in xylem and fibre cells, in which lignification is initiated in the cell corners and middle lamella and then proceeds inwards following the deposition of polysaccharides, is currently hard to say. It is also possible that other sources of apoplastic $H_2O_2$ may be present with different distribution patterns within the wall [110].

Dirigent proteins (DPs) were first reported to be involved in directing the stereoselective coupling of two laccase-generated monolignol radicals in lignan biosynthesis, and subsequently invoked as components of the lignin polymerization machinery [19,114]. Since then, their involvement in the latter has been hotly debated. Many DPs are induced in response to biotic and abiotic stress, and it has been assumed, although not conclusively proven, that this might represent involvement in the biosynthesis of defensive lignans and/or strengthening of the cell wall via lignin deposition (reviewed in [115]). The best genetic evidence for involvement of a DP in lignin, as opposed to lignan, biosynthesis, again comes from the study of the development of the CS. The loss of function of Enhanced Suberin 1 (ESB1), a dirigent domain-containing protein, results in spatially incorrect lignin deposition in the CS, but does not actually prevent lignin accumulation [116]. The ESB1 protein localizes to the endodermis in a manner consistent with development of the CS, and this localization is disrupted on the loss of function of the scaffold protein CASP1 [116]. It is tempting to speculate that CASP1 brings ESPB1 into the complex with NOX (see above) in order to facilitate monolignol oligomerization through targeted phenoxy radical coupling (figure 5c). However, the precise biochemical function of ESB1 is unknown; the protein does not confer stereoselectivity on the laccase-mediated oxidation/dimerization of coniferyl alcohol. However, this is not inconsistent with the model of Davin & Lewis [19], in which arrays of dirigent 'sites' could allow for regioselective (but not necessarily stereoselective) 8-O-4' coupling to initiate lignification. Additional DPs appear to be expressed in the endodermis of *A. thaliana* and might complement the activity of ESB1 [116]. It is also interesting that the loss of function of pinoresinol reductase 1, the enzyme catalysing the reaction subsequent to stereoselective monolignol coupling in lignan synthesis, results in a change in lignin distribution, but not content or composition, in

*A. thaliana* [117]. This provides further support for a model implicating components of lignan synthesis to targeting of lignin distribution within the cell wall.

Localized complexes containing laccases or peroxidases might explain the sites of lignin initiation. Another emerging question is whether there is the need for some type of initiator molecule to start the lignin chain. Early biochemical studies identified cell wall-esterified ferulate resides as potential nucleation site for lignin [118], but this has been hard to prove genetically. The recent demonstration of tricin as a component of lignin in grasses, and its apparently exclusive 4-O-β linkage to monolignols, has been interpreted as indicating that tricin is an initiator of lignification in monocots [27]. The loss of function of an essential cytochrome P450 enzyme required for tricin biosynthesis resulted in tricin-deficient rice plants with reduced levels of lignin which was also less condensed [119]. The tricin was partially replaced with its precursor, the flavanone naringenin, which can couple with monolignol through the B-ring by both 4-O-β and 3'β coupling [119]. This finding, and the fact that *Z. mays* plants that do not produce flavonoids as a result of lack of function of chalcone synthase appear to have increased rather than reduced levels of lignin [120], suggests that tricin is not an obligatory initiator of lignification in grasses, and that its B-ring substitution pattern, which is identical to that of the S-lignin monomer, is not essential for its incorporation into lignin.

The model in figure 5c presents a hypothetical scheme in which an initiator molecule (shown as tricin) couples with a monolignol(s) to yield an initiator dimer or oligomer (ID/O). The ID/O then diffuses to the sites where peroxidase catalyses bulk lignin polymerization. In the CS, laccase is not required, and the initiation sites may be associated with NOX through MASPs and DPs.

# 7. Conclusion: still more questions than answers?

Our new understanding of the flexibility of lignin biosynthesis, the bi-functionality of some of the biosynthetic enzymes, the potential involvement of parallel routes in the pre-G portion of the pathway and the emerging evidence for spatial control mechanisms for the initiation of lignification raises as many questions as it answers. Addressing the following questions would appear to be fruitful for future studies.

1. Did monolignol biosynthesis evolve from a mechanism designed to protect cells from stress-induced ROS?
2. Does hydrogen peroxide act as a cytosolic signal to coordinate monolignol biosynthesis and polymerization?
3. Does the acids pathway operate for biosynthesis of all monolignols (H, G and S)?
4. Are the SS and the acids pathway to caffeic acid under separate environmental and developmental control? Detailed labelling studies (e.g. [121]) may help answer this question.
5. Is TAL preferentially expressed in cells making predominantly S-lignin (e.g. fibre cells)?
6. Is the pool of coumarate generated from TAL biochemically distinct from that generated via PAL/C4H?
7. What is the biochemical function of the complex between C4H, C3'H and F5H on the ER?

8. Does ACBP2 help localize CSE at the plasma membrane as a site for monolignol biosynthesis?

9. Does CSE localize to the plasma membrane to facilitate directional monolignol transport and link monolignol synthesis to antioxidant activity for lipid repair?

10. Does the activation of CCR via interaction with Rac GTPase only occur in the case of defence-induced lignification?

11. Is monolignol transport predominantly a passive process? Or, does biochemical evidence for ATP-dependent monolignol transport processes [122] mean that active transport is quantitatively important?

12. Is the glycan portion of laccases important for targeting the enzymes to lignin initiation sites?

13. What is the physical nature of the sites within the apoplast where initiation of lignification occurs, and is a specific chemical initiator(s) of lignification made in the cytosol or apoplast?

14. Are monolignols always present and mobile in the apoplast, or are they specifically targeted across the plasma membrane to sites of lignin initiation?

15. What are the differences in the lignin initiation process in different cell types?

Data accessibility. This article has no additional data.

Authors' contributions. R.A.D. and J.B. co-wrote the paper. R.A.D. designed the figures and J.B. edited them.

Competing interests. We declare we have no competing interests.

Funding. Our research on lignin biosynthesis and engineering is supported by the Center for Bioenergy Innovation, which is a US DOE Bioenergy Research Center supported by the Office of Biological and Environmental Research in the DOE Office of Science, and by the US National Science Foundation Integrated Organismal Systems program (grant no. 1456286).

Acknowledgements. R.A.D. acknowledges a Faculty Fellowship from the Hagler Institute for Advanced Study at Texas A&M University, during which period this article was written.

## Authors' profile

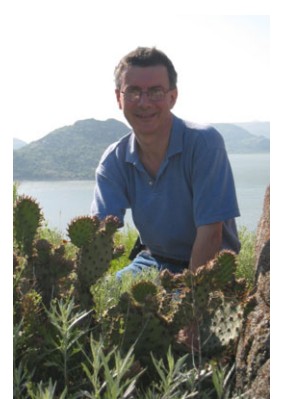

**Richard A. Dixon** is a faculty fellow of the Hagler Institute for Advanced Study, eminent scholar in residence, visiting professor in the College of Science and Timothy C. Hall-Heep Distinguished Faculty Chair at Texas A&M University, College Station, TX, USA; and distinguished research professor in the Department of Biological Sciences and associate director of the BioDiscovery Institute at the University of North Texas, Denton, TX, USA.

Richard's research centres on the biochemistry, molecular biology and metabolic engineering of plant natural product pathways and their implications for agriculture, energy and human health. His recent studies focus on understanding the biosynthetic pathways leading to lignin and condensed tannins, and his research has been instrumental in the development of commercial reduced lignin alfalfa with improved forage quality and harvest flexibility.

Richard serves on the science advisory boards of several major research institutes, including VIB Gent and the Agricultural Biotechnology Research Center at Academia Sinica, Taipei, Taiwan. He was awarded the Groupe Polyphenols Scientific Prize (2012), the Shang-Fa Yang Memorial Lectureship from Academia Sinica, Taiwan (2012) and the Phytochemical Pioneer Award from the Phytochemical Society of North America (2015). He was elected member of the US National Academy of Sciences (2007), and fellow of the Royal Society of London (2018). He is also an elected fellow of the American Association for the Advancement of Science (2003), the National Academy of Inventors (2014) and the American Society of Plant Biologists (2018).

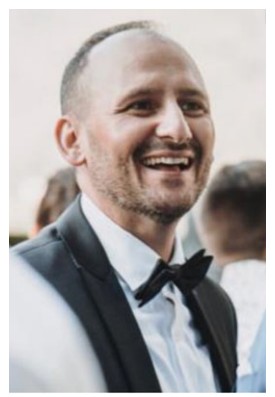

**Jaime Barros** completed his undergraduate studies in Forestry Engineering at the University of Santiago de Compostela and commenced doctoral studies in Plant Breeding and Genetics at the University of Vigo, where he received his Dr Phil in 2012. He pursued postdoctoral research training in plant molecular biology and biochemistry at the Umeå Plant Science Center and the University of North Texas. Jaime was a Barrie Foundation Fellowship recipient in 2012 and a GAIN-Postdoctoral Fellow in 2015. His research interests focus on the biology of wood formation in higher plants. In particular, he has recently been pioneering work on the biochemical reactions leading to lignin biosynthesis in monocotyledonous plants. In 2019, he was awarded the CBI-Early Career Development Award funded by the US Department of Energy and is a Visiting Fellow at Oak Ridge National Laboratory, where he has used multi-omic approaches to delineate metabolic responses associated with lignin biosynthesis perturbations.

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
