## [Reviewer comments · Open Biology]

Review History

RSOB-19-0215.R0 (Original submission)

Review form: Reviewer 1

Recommendation

Accept as is

Do you have any ethical concerns with this paper?

No

Comments to the Author

Dixon and Barros have written a wonderful review of the complexities of lignin biosynthesis. Biochemical deciphering of the pathway has a long history, where ingenious discoveries have many times rearranged the thinking of how the pathway works. For example, the hiding activity of C3H was found to be buried in a transesterification reaction (the shikimate shunt), central for the lignin monomer biosynthesis were its inhibition (by down regulation of HCT) caused severe defects. After absorbing this fact, the scientific community was given a new surprise by reports of two alternative and direct reactions for the 3-hydroxylation, by a C4H-C3H membrane associated complex, or by a soluble C3H that turned out to be a peroxidase.

The history, present state and unsolved questions of lignin biosynthesis are described here in an exciting and insightful way.

Review form: Reviewer 2

Recommendation

Accept with minor revision (please list in comments)

Do you have any ethical concerns with this paper?

No

Comments to the Author

This is a well-written, comprehensive and inspiring perspective paper concerning the biochemistry and cell biology aspects of lignin biosynthesis. The authors are one of the most active research groups in this particular research area, and discuss timely and important topics within the current manuscript. Given that the topics covered in this manuscript are highly relevant to recent biotechnology efforts aiming to improve the quality of plant biomass for sustainable utilizations, I believe this manuscript will receive much attention from both plant biology and biotechnology communities. I have some minor suggestions that may help to improve the present version.

P.4 Line 101: The authors may consider citing also the very recently study reporting that hydroxystilbenes, more specifically their glycosides, are also authentic lignin monomers in *Picea abies* bark tissues (Rencoret J et al., *Plant Physiol*, 180, 1310-1321, 2019).

P.6 Line 151: The essential function of the (C3'H)/HCT-mediated SS has been demonstrated not only in angiosperms i.e., dicots and monocots, but also in gymnosperms (Wagner A. et al. *PNAS*, 104, 11856-11861, 2007).

P.8 Line 218: Such less (and unique) impacts of C3'H downregulations on lignin deposition and composition in monocots compared to dicots are also well investigated in a recent study with rice (Takeda et al., *Plant J.*, 95, 796-811, 2018).

P.8 Line 228: "endoplasmic reticulum" -> "endoplasmic reticulum (ER)". The authors should consistently use (or not use) this abbreviation in the subsequent texts.

P.11 Line 291 (and throughout this manuscript): If the authors define the abbreviation "A. thaliana lysophospholipase 2 (CSE/LPL)" (or CSE/LPL2??), they should consistently use this abbreviation in the subsequent texts.

P.12 Line 317: It may be informative for readers to note that the specific protein class of the NADPH oxidase (NOX) enzymes considered as ROS sources for lignin biosynthesis are the respiratory burst oxidase homologue (RBOH) proteins.

P.12 Line 330: "lysophospholipase C/CSE" -> "lysophospholipase 2/CSE" or "CSE/LPL" (or "CSE/LPL2"??)

P.14 Line 385: "...transport in living (e.g. the CS)" -> "...transport in living (e.g. the Casparian strips, CS, see below)". The abbreviation for CS appears for the first time here.

P.15 Line 425 or around: The authors may consider commenting on the recently published study which reported that CASPs (or CASP-like proteins, CASPLs) also critically function for pathogen-induced lignification in Arabidopsis leaves (Lee et al., EMBO J., e101948, 2019).

P.17 Line 467-469: "the fact that corn plants that do not produce flavonoids as a result of lack of function of chalcone synthase..." This statement needs citation. I think it is by Eloy et al., Plant Physiol., 173, 998-1016, 2017?

Figure 1 (Image): (a) the monolignol unit connecting the branching chain via 5-O-4 linkage should be "G", not "S" unit. (c) The double bond in the H monolignol should be between C α and C β , now incorrectly between C β and C γ .

Figure 2 (Image): (a) The C α =C β double bond in coniferaldehyde is missing. The double bond in the H monolignol should be between C α and C β , now incorrectly between C β and C γ (same as in Figure 1). (c) "G > 80%" and "G > 75%" (in alfalfa and Arabidopsis composition) -> "G \approx 80%" and "G \approx 75%"??

Figure 5. (a) and (b): Why do the authors show the WRKY mutant stems here? Should be wild-type stems, or need some explanations about WRKY in the main text or in legend?

Figure 1 (legend): "Coniferyl alcohol (C)" -> "Coniferyl alcohol (G)"

Figure 2 (legend): "5, hydroxycinnamate CoA ligase;" -> "5, 4-hydroxycinnamate (or 4-coumarate) CoA ligase (4CL);", "9, ferulic acid/coniferaldehyde 5-hydroxylase" -> "9, ferulic acid/coniferaldehyde 5-hydroxylase (F5H)"

Figure 5 (legend): "in the CS along with DPs" -> "in the Casparian strips (CS) along with dirigent proteins (DPs)" Also, it would be nice to note that MASPs here include CASPs (or CASP-like proteins, CASPLs) mentioned in the main text?

Review form: Reviewer 3

Recommendation

Major revision is needed (please make suggestions in comments)

Do you have any ethical concerns with this paper?

No

Comments to the Author

In the ms submitted, the authors have reviewed lignin biosynthesis, and propose several hypothetical models for still unknown questions. This is a nicely written review that gives fresh ideas for further investigation. The review is based mostly on studies of Arabidopsis, Medicago and grasses, with only some citations to the work conducted in woody plants (mostly Populus). Since woody plants contain most of the lignin due to the huge biomass, it would have been good to additionally discuss original research papers related to them. Especially studies about coniferous (gymnosperm) species are not cited at all. The reader should be informed about this in the beginning of the review.

I have following comments:

Thickening of the secondary cell wall is very prominent in tracheary elements, tracheids and fibers, the main lignifying cell types in xylem. A Casparian strip in the root – an excellent model for lignification studies - lacks this cell wall thickening. Thickening of the wall is an important point that needs to be taken into account when designing models. Fig. 5 proposes a model where peroxidase is in a module with dirigent protein and MASP, close to NADPH oxidase. This is based on an idea that has been observed in the Casparian strip. It is a nice idea, but can it be somehow realistic in a thickening secondary cell wall of tracheary elements and fibers during development? From the literature, it is known that lignification follows polysaccharide synthesis. Polysaccharides are produced first, making cell wall thick, and after polysaccharide synthesis has proceeded to a certain stage, lignification initiates farthest away from the membrane (in cell corners and the middle lamella), and the proceeds inwards, but always behind polysaccharide synthesis. If an oxidative enzyme, e.g. peroxidase, is located next to plasma membrane, it could participate only when lignification is almost finished, and not in the main lignification of the thick secondary wall. Similarly, if NADPH oxidase is the only source of apoplastic H₂O₂, could that diffuse around the apoplast to the site where that is needed without being reacted closer to the site of its generation? In addition to NADPH oxidase, there are other potential enzymatic sources for apoplastic H₂O₂ (e.g. peroxidases, several types of oxidases, e.g. Kärkönen and Kuchitsu, 2015, *Phytochemistry*). Some of these could equally well be involved in H₂O₂ production needed for lignification (even in addition to NADPH oxidase) – with more broad distribution in the wall.

l. 408-409. There are several older papers on trees that lignification is initiated in cell corners and middle lamella – please, cite some older papers. It is interesting to notice that ectopic lignin formation is also initiated in cell corners. It would be good to have a figure of developmental lignification next to this in Fig. 5 (or citation to a figure showing this).

l. 331-332 and Figure 4 legend: “Hydrogen peroxide is generated in nearly all intracellular compartments, with a major source being the plasma membrane-associated NADPH oxidase (NOX) with its catalytic site in the apoplast.”

NADPH oxidases may be a major source for apoplastic ROS production, but since ROS are generated e.g. in chloroplasts and peroxisomes, NADPH oxidase may not be the main cellular source for H₂O₂ generation in photosynthesizing cells. NADPH oxidases transfer electrons across the plasma membrane from cytoplasmic NADPH to molecular oxygen to produce superoxide in the apoplastic side of the membrane. Thus, product generation occurs in the apoplast, but electrons originate from NADPH that is in the cytoplasmic side, and are transferred via the heme groups in the transmembrane areas (Sagi and Fluhr 2006). Hence, catalytic site locates in the cytoplasmic side and through the membrane, and product is generated in the apoplastic side.

l. 329-344. This chapter discusses the need to transport H₂O₂ from the apoplast to cytosol to activate lysophospholipase C/CSE. Authors need to get familiar with papers about cytosolic ROS generation and antioxidants. There are multiple sources of cytoplasmic H₂O₂ generation, and efficient antioxidant systems (additionally to APX) that keep the redox balance of the cytosol highly reduced (e.g. Noctor and Foyer, 2016, *Plant Cell Environm*; Smirnov and Arnaud, 2019, *New Phytol*). However, ROS signalling between different organs and compartments is known to be present (e.g. Waszczak et al., 2018, *Annu Rev Plant Biol*). It has to be kept in mind that there needs to be a concentration gradient between the sides of plasma membrane with higher concentration in the apoplast so that diffusion inwards is occurring. This gradient probably occurs in the defense response with RBOHs activated. Is the gradient enough during

developmental lignification when H₂O₂ is consumed by peroxidases in the apoplastic space? Please, consider this knowledge and rewrite this chapter.

It is not uniform how plant species' names are written (sometimes only in English, sometimes only in latin). Please, uniform the naming.

Minor comments:

- l. 26 ... highly controlled polymerization procedure... The idea of "random polymerization" is not equal to highly controlled procedure - please rephrase.
- l. 33-34: H₂O₂ is an oxidant for peroxidases. Laccases that also oxidise monolignols to radicals use molecular oxygen. Please, rephrase.
- l. 65. Extracellular - Please change to "apoplastic" to be uniform with the rest of the text.
- l. 279: .. the others being CSE and CCR. - please add references in this sentence.
- l. 343: ... in the cell walls of knock-out lines of superoxide dismutase.
- l. 385: CS - write open when first mentioned.
- l. 905-906: coniferyl alcohol (G). The order of the compounds in the figure legend could be the same than that in the figure (starting with H).
- l. 956: ("oxidation/polymerization via peroxidases"): Peroxidase oxidizes monolignol to monolignol radical - polymerization does not require H₂O₂ or the peroxidase enzyme anymore - > remove "polymerization".

Fig. 1a: coloured (red) lines are not very visible.

Fig. 2: 5HG-unit (to be uniform with the text)

Decision letter (RSOB-19-0215.R0)

18-Oct-2019

Dear Dr Dixon

We are pleased to inform you that your manuscript RSOB-19-0215 entitled "Lignin biosynthesis - old roads revisited and new roads explored" has been accepted by the Editor for publication in Open Biology. The reviewer(s) have recommended publication, but also suggest some minor revisions to your manuscript. Therefore, we invite you to respond to the reviewer(s)' comments and revise your manuscript.

Please submit the revised version of your manuscript within 7 days. If you do not think you will be able to meet this date please let us know immediately and we can extend this deadline for you.

To revise your manuscript, log into <https://mc.manuscriptcentral.com/rsob> and enter your Author Centre, where you will find your manuscript title listed under "Manuscripts with

Decisions." Under "Actions," click on "Create a Revision." Your manuscript number has been appended to denote a revision.

- 1) A text file of the manuscript (doc, txt, rtf or tex), including the references, tables (including captions) and figure captions. Please remove any tracked changes from the text before submission. PDF files are not an accepted format for the "Main Document".
- 2) A separate electronic file of each figure (tiff, EPS or print-quality PDF preferred). The format should be produced directly from original creation package, or original software format. Please note that PowerPoint files are not accepted.
- 3) Electronic supplementary material: this should be contained in a separate file from the main text and meet our ESM criteria (see <http://royalsocietypublishing.org/instructions-authors#question5>). All supplementary materials accompanying an accepted article will be treated as in their final form. They will be published alongside the paper on the journal website and posted on the online figshare repository. Files on figshare will be made available approximately one week before the accompanying article so that the supplementary material can be attributed a unique DOI.

Online supplementary material will also carry the title and description provided during submission, so please ensure these are accurate and informative. Note that the Royal Society will not edit or typeset supplementary material and it will be hosted as provided. Please ensure that the supplementary material includes the paper details (authors, title, journal name, article DOI). Your article DOI will be 10.1098/rsob.2016[last 4 digits of e.g. 10.1098/rsob.20160049].

- 4) A media summary: a short non-technical summary (up to 100 words) of the key findings/importance of your manuscript. Please try to write in simple English, avoid jargon, explain the importance of the topic, outline the main implications and describe why this topic is newsworthy.

Images

Data-Sharing

It is a condition of publication that data supporting your paper are made available. Data should be made available either in the electronic supplementary material or through an appropriate repository. Details of how to access data should be included in your paper. Please see <http://royalsocietypublishing.org/site/authors/policy.xhtml#question6> for more details.

Data accessibility section

Sincerely,

The Open Biology Team

mailto:openbiology@royalsociety.org

Reviewer(s)' Comments to Author:

Referee: 1

Comments to the Author(s)

Dixon and Barros have written a wonderful review of the complexities of lignin biosynthesis. Biochemical deciphering of the pathway has a long history, where ingenious discoveries have many times rearranged the thinking of how the pathway works. For example, the hiding activity of C3H was found to be buried in a transesterification reaction (the shikimate shunt), central for the lignin monomer biosynthesis were its inhibition (by down regulation of HCT) caused severe defects. After absorbing this fact, the scientific community was given a new surprise by reports of two alternative and direct reactions for the 3-hydroxylation, by a C4H-C3H membrane associated complex, or by a soluble C3H that turned out to be a peroxidase.

The history, present state and unsolved questions of lignin biosynthesis are described here in an exciting and insightful way.

Referee: 2

Comments to the Author(s)

This is a well-written, comprehensive and inspiring perspective paper concerning the biochemistry and cell biology aspects of lignin biosynthesis. The authors are one of the most active research groups in this particular research area, and discuss timely and important topics within the current manuscript. Given that the topics covered in this manuscript are highly relevant to recent biotechnology efforts aiming to improve the quality of plant biomass for sustainable utilizations, I believe this manuscript will receive much attention from both plant biology and biotechnology communities. I have some minor suggestions that may help to improve the present version.

P.4 Line 101: The authors may consider citing also the very recently study reporting that hydroxystilbenes, more specifically their glycosides, are also authentic lignin monomers in *Picea abies* bark tissues (Rencoret J et al., *Plant Physiol*, 180, 1310-1321, 2019).

P.6 Line 151: The essential function of the (C3'H/)HCT-mediated SS has been demonstrated not only in angiosperms i.e., dicots and monocots, but also in gymnosperms (Wagner A. et al. PNAS, 104, 11856-11861, 2007).

P.8 Line 218: Such less (and unique) impacts of C3'H downregulations on lignin deposition and composition in monocots compared to dicots are also well investigated in a recent study with rice (Takeda et al., Plant J., 95, 796-811, 2018).

P.8 Line 228: "endoplasmic reticulum" -> "endoplasmic reticulum (ER)". The authors should consistently use (or not use) this abbreviation in the subsequent texts.

P.11 Line 291 (and throughout this manuscript): If the authors define the abbreviation "A. thaliana lysophospholipase 2 (CSE/LPL)" (or CSE/LPL2??), they should consistently use this abbreviation in the subsequent texts.

P.12 Line 317: It may be informative for readers to note that the specific protein class of the NADPH oxidase (NOX) enzymes considered as ROS sources for lignin biosynthesis are the respiratory burst oxidase homologue (RBOH) proteins.

P.12 Line 330: "lysophospholipase C/CSE" -> "lysophospholipase 2/CSE" or "CSE/LPL" (or "CSE/LPL2")??

P.14 Line 385: "...transport in living (e.g. the CS)" -> "...transport in living (e.g. the Casparian strips, CS, see below)". The abbreviation for CS appears for the first time here.

P.15 Line 425 or around: The authors may consider commenting on the recently published study which reported that CASPs (or CASP-like proteins, CASPLs) also critically function for pathogen-induced lignification in Arabidopsis leaves (Lee et al., EMBO J., e101948, 2019).

P.17 Line 467-469: "the fact that corn plants that do not produce flavonoids as a result of lack of function of chalcone synthase..." This statement needs citation. I think it is by Eloy et al., Plant Physiol., 173, 998-1016, 2017?

Figure 1 (Image): (a) the monolignol unit connecting the branching chain via 5-O-4 linkage should be "G", not "S" unit. (c) The double bond in the H monolignol should be between Ca and C β , now incorrectly between C β and C?

Figure 2 (Image): (a) The Ca=C β double bond in coniferaldehyde is missing. The double bond in the H monolignol should be between Ca and C β , now incorrectly between C β and C? (same as in Figure 1). (c) "G > 80%" and "G > 75%" (in alfalfa and Arabidopsis composition) -> "G ~ 80%" and "G ~ 75%"??

Figure 5. (a) and (b): Why do the authors show the WRKY mutant stems here? Should be wild-type stems, or need some explanations about WRKY in the main text or in legend?

Figure 1 (legend): "Coniferyl alcohol (C)" -> "Coniferyl alcohol (G)"

Figure 2 (legend): "5, hydroxycinnamate CoA ligase;" -> "5, 4-hydroxycinnamate (or 4-coumarate) CoA ligase (4CL);", "9, ferulic acid/coniferaldehyde 5-hydroxylase" -> "9, ferulic acid/coniferaldehyde 5-hydroxylase (F5H)"

Figure 5 (legend): “in the CS along with DPs” -> “in the Casparian strips (CS) along with dirigent proteins (DPs)” Also, it would be nice to note that MASPs here include CASPs (or CASP-like proteins, CASPLs) mentioned in the main text?

Referee: 3

Comments to the Author(s)

In the ms submitted, the authors have reviewed lignin biosynthesis, and propose several hypothetical models for still unknown questions. This is a nicely written review that gives fresh ideas for further investigation. The review is based mostly on studies of Arabidopsis, Medicago and grasses, with only some citations to the work conducted in woody plants (mostly Populus). Since woody plants contain most of the lignin due to the huge biomass, it would have been good to additionally discuss original research papers related to them. Especially studies about coniferous (gymnosperm) species are not cited at all. The reader should be informed about this in the beginning of the review.

I have following comments:

Thickening of the secondary cell wall is very prominent in tracheary elements, tracheids and fibers, the main lignifying cell types in xylem. A Casparian strip in the root – an excellent model for lignification studies - lacks this cell wall thickening. Thickening of the wall is an important point that needs to be taken into account when designing models. Fig. 5 proposes a model where peroxidase is in a module with dirigent protein and MASP, close to NADPH oxidase. This is based on an idea that has been observed in the Casparian strip. It is a nice idea, but can it be somehow realistic in a thickening secondary cell wall of tracheary elements and fibers during development? From the literature, it is known that lignification follows polysaccharide synthesis. Polysaccharides are produced first, making cell wall thick, and after polysaccharide synthesis has proceeded to a certain stage, lignification initiates farthest away from the membrane (in cell corners and the middle lamella), and the proceeds inwards, but always behind polysaccharide synthesis. If an oxidative enzyme, e.g. peroxidase, is located next to plasma membrane, it could participate only when lignification is almost finished, and not in the main lignification of the thick secondary wall. Similarly, if NADPH oxidase is the only source of apoplastic H₂O₂, could that diffuse around the apoplast to the site where that is needed without being reacted closer to the site of its generation? In addition to NADPH oxidase, there are other potential enzymatic sources for apoplastic H₂O₂ (e.g. peroxidases, several types of oxidases, e.g. Kärkönen and Kuchitsu, 2015, Phytochemistry). Some of these could equally well be involved in H₂O₂ production needed for lignification (even in addition to NADPH oxidase) – with more broad distribution in the wall.

l. 408-409. There are several older papers on trees that lignification is initiated in cell corners and middle lamella – please, cite some older papers. It is interesting to notice that ectopic lignin formation is also initiated in cell corners. It would be good to have a figure of developmental lignification next to this in Fig. 5 (or citation to a figure showing this).

l. 331-332 and Figure 4 legend: “Hydrogen peroxide is generated in nearly all intracellular compartments, with a major source being the plasma membrane-associated NADPH oxidase (NOX) with its catalytic site in the apoplast.”

NADPH oxidases may be a major source for apoplastic ROS production, but since ROS are generated e.g. in chloroplasts and peroxisomes, NADPH oxidase may not be the main cellular source for H₂O₂ generation in photosynthesizing cells. NADPH oxidases transfer electrons across the plasma membrane from cytoplasmic NADPH to molecular oxygen to produce superoxide in the apoplastic side of the membrane. Thus, product generation occurs in the

apoplast, but electrons originate from NADPH that is in the cytoplasmic side, and are transferred via the heme groups in the transmembrane areas (Sagi and Fluhr 2006). Hence, catalytic site locates in the cytoplasmic side and through the membrane, and product is generated in the apoplastic side.

l. 329-344. This chapter discusses the need to transport H₂O₂ from the apoplast to cytosol to activate lysophospholipase C/CSE. Authors need to get familiar with papers about cytosolic ROS generation and antioxidants. There are multiple sources of cytoplasmic H₂O₂ generation, and efficient antioxidant systems (additionally to APX) that keep the redox balance of the cytosol highly reduced (e.g. Noctor and Foyer, 2016, *Plant Cell Environm*; Smirnov and Arnaud, 2019, *New Phytol*). However, ROS signalling between different organs and compartments is known to be present (e.g. Waszczak et al., 2018, *Annu Rev Plant Biol*). It has to be kept in mind that there needs to be a concentration gradient between the sides of plasma membrane with higher concentration in the apoplast so that diffusion inwards is occurring. This gradient probably occurs in the defense response with RBOHs activated. Is the gradient enough during developmental lignification when H₂O₂ is consumed by peroxidases in the apoplastic space? Please, consider this knowledge and rewrite this chapter.

It is not uniform how plant species' names are written (sometimes only in English, sometimes only in latin). Please, uniform the naming.

Minor comments:

- l. 26 ... highly controlled polymerization procedure... The idea of "random polymerization" is not equal to highly controlled procedure - please rephrase.
- l. 33-34: H₂O₂ is an oxidant for peroxidases. Laccases that also oxidise monolignols to radicals use molecular oxygen. Please, rephrase.
- l. 65. Extracellular - Please change to "apoplastic" to be uniform with the rest of the text.
- l. 279: .. the others being CSE and CCR. - please add references in this sentence.
- l. 343: ... in the cell walls of knock-out lines of superoxide dismutase.
- l. 385: CS - write open when first mentioned.
- l. 905-906: coniferyl alcohol (G). The order of the compounds in the figure legend could be the same than that in the figure (starting with H).
- l. 956: ("oxidation/polymerization via peroxidases"): Peroxidase oxidizes monolignol to monolignol radical - polymerization does not require H₂O₂ or the peroxidase enzyme anymore - > remove "polymerization".

Fig. 1a: coloured (red) lines are not very visible.

Fig. 2: 5HG-unit (to be uniform with the text)

Author's Response to Decision Letter for (RSOB-19-0215.R0)

See Appendix A.

Decision letter (RSOB-19-0215.R1)

30-Oct-2019

Dear Dr Dixon

We are pleased to inform you that your manuscript entitled "Lignin biosynthesis - old roads revisited and new roads explored" has been accepted by the Editor for publication in Open Biology.

Sincerely,

The Open Biology Team
mailto: openbiology@royalsociety.org

Appendix A

Response to reviews

Referee: 1

Comments to the Author(s)

Dixon and Barros have written a wonderful review of the complexities of lignin biosynthesis. Biochemical deciphering of the pathway has a long history, where ingenious discoveries have many times rearranged the thinking of how the pathway works. For example, the hiding activity of C3H was found to be buried in a transesterification reaction (the shikimate shunt), central for the lignin monomer biosynthesis were its inhibition (by down regulation of HCT) caused severe defects. After absorbing this fact, the scientific community was given a new surprise by reports of two alternative and direct reactions for the 3-hydroxylation, by a C4H-C3H membrane associated complex, or by a soluble C3H that turned out to be a peroxidase.

The history, present state and unsolved questions of lignin biosynthesis are described here in an exciting and insightful way.

Response. Thank you for the positive response.

Referee: 2

Comments to the Author(s)

This is a well-written, comprehensive and inspiring perspective paper concerning the biochemistry and cell biology aspects of lignin biosynthesis. The authors are one of the most active research groups in this particular research area, and discuss timely and important topics within the current manuscript. Given that the topics covered in this manuscript are highly relevant to recent biotechnology efforts aiming to improve the quality of plant biomass for sustainable utilizations, I believe this manuscript will receive much attention from both plant biology and biotechnology communities. I have some minor suggestions that may help to improve the present version.

Response 1. Thank you for the positive response, and for the suggestions below, especially those concerning relevant papers that we had missed. We hope we have fully addressed the omissions in the revised MS.

P.4 Line 101: The authors may consider citing also the very recently study reporting that hydroxystilbenes, more specifically their glycosides, are also authentic lignin monomers in *Picea abies* bark tissues (Rencoret J et al., *Plant Physiol*, 180, 1310-1321, 2019).

Response 2. Done.

P.6 Line 151: The essential function of the (C3'H)/HCT-mediated SS has been demonstrated not only in angiosperms i.e., dicots and monocots, but also in gymnosperms (Wagner A. et al. *PNAS*, 104, 11856-11861, 2007).

Response 3. Yes- we have now added the citation.

P.8 Line 218: Such less (and unique) impacts of C3'H downregulations on lignin deposition and composition in monocots compared to dicots are also well investigated in a recent study with rice (Takeda et al., Plant J., 95, 796-811, 2018).

Response 4. We agree, although this paper does show fairly strong increases in H lignin in C3'H-RNAi rice, and a very strong growth defect on total ablation of C3'H by gene editing. Because comparing these data (i.e. different effects of total loss of function and RNAi-mediated silencing) to what has been reported in other monocots (only by RNAi) may be confusing to someone not familiar with this field, we have chosen to not cite the reference, but we do qualify the sentence to indicate that the less severe effects are seen in "some" monocots (i.e. there are exceptions). The point of this discussion was to set out the argument for why we looked for the alternative coumarate hydroxylation pathway.

P.8 Line 228: "endoplasmic reticulum" -> "endoplasmic reticulum (ER)". The authors should consistently use (or not use) this abbreviation in the subsequent texts.

Response 5. Done. As with other abbreviations referred to below, we now consistently define on first appearance and then use the abbreviation.

P.11 Line 291 (and throughout this manuscript): If the authors define the abbreviation "A. thaliana lysophospholipase 2 (CSE/LPL)" (or CSE/LPL2??), they should consistently use this abbreviation in the subsequent texts.

Response 6. Done

P.12 Line 317: It may be informative for readers to note that the specific protein class of the NADPH oxidase (NOX) enzymes considered as ROS sources for lignin biosynthesis are the respiratory burst oxidase homologue (RBOH) proteins.

Response 7. Done

P.12 Line 330: "lysophospholipase C/CSE" -> "lysophospholipase 2/CSE" or "CSE/LPL" (or "CSE/LPL2")??

Response 8. Corrected the typo

P.14 Line 385: "...transport in living (e.g. the CS)" -> "...transport in living (e.g. the Casparian strips, CS, see below)". The abbreviation for CS appears for the first time here.

Response 9. Now defined Casparian strip on first appearance, and used abbreviation subsequently.

P.15 Line 425 or around: The authors may consider commenting on the recently published study which reported that CASPs (or CASP-like proteins, CASPLs) also critically function for pathogen-induced lignification in Arabidopsis leaves (Lee et al., EMBO J., e101948, 2019).

Response 10. Thanks for pointing this out. This is an important piece of work that further supports our discussions, and is now cited.

P.17 Line 467-469: "the fact that corn plants that do not produce flavonoids as a result of lack of function of chalcone synthase..." This statement needs citation. I think it is by Eloy et al., Plant Physiol., 173, 998-1016, 2017?

Response 11. Citation added.

Figure 1 (Image): (a) the monolignol unit connecting the branching chain via 5-O-4 linkage should be "G", not "S" unit. (c) The double bond in the H monolignol should be between C α and C β , now incorrectly between C β and C γ .

Response 12. Thanks for spotting these errors. They have been corrected.

Figure 2 (Image): (a) The C α =C β double bond in coniferaldehyde is missing. The double bond in the H monolignol should be between C α and C β , now incorrectly between C β and C γ (same as in Figure 1). (c) "G > 80%" and "G > 75%" (in alfalfa and Arabidopsis composition) -> "G \approx 80%" and "G \approx 75%"??

Response 13. Done

Figure 5. (a) and (b): Why do the authors show the WRKY mutant stems here? Should be wild-type stems, or need some explanations about WRKY in the main text or in legend?

Response 14. This was fully explained in the legend. WRKY knock-down cause ectopic lignification in the pith cells, and, in the early stages, this nicely shows the initiation of lignification in the cell corners.

Figure 1 (legend): "Coniferyl alcohol (C)" -> "Coniferyl alcohol (G)"

Response 15. Typo corrected.

Figure 2 (legend): "5, hydroxycinnamate CoA ligase;" -> "5, 4-hydroxycinnamate (or 4-coumarate) CoA ligase (4CL);", "9, ferulic acid/coniferaldehyde 5-hydroxylase" -> "9, ferulic acid/coniferaldehyde 5-hydroxylase (F5H)"

Response 16. Thanks for spotting this. Abbreviations added.

Figure 5 (legend): "in the CS along with DPs" -> "in the Casparian strips (CS) along with dirigent proteins (DPs)" Also, it would be nice to note that MASPs here include CASPs (or CASP-like proteins, CASPLs) mentioned in the main text?

Response 17. Good idea- we have included this in both the text and revised Figure 5.

Referee: 3

Comments to the Author(s)

In the ms submitted, the authors have reviewed lignin biosynthesis, and propose several hypothetical models for still unknown questions. This is a nicely written review that gives fresh ideas for further investigation. The review is based mostly on studies of Arabidopsis, Medicago and grasses, with only some citations to the work conducted in woody plants (mostly Populus). Since woody plants contain

most of the lignin due to the huge biomass, it would have been good to additionally discuss original research papers related to them. Especially studies about coniferous (gymnosperm) species are not cited at all. The reader should be informed about this in the beginning of the review.

Response 1. We agree that this review mainly deals with studies in model species, and have now made this fact clear in the summary.

I have following comments:

Thickening of the secondary cell wall is very prominent in tracheary elements, tracheids and fibers, the main lignifying cell types in xylem. A Casparian strip in the root – an excellent model for lignification studies - lacks this cell wall thickening. Thickening of the wall is an important point that needs to be taken into account when designing models. Fig. 5 proposes a model where peroxidase is in a module with dirigent protein and MASP, close to NADPH oxidase. This is based on an idea that has been observed in the Casparian strip. It is a nice idea, but can it be somehow realistic in a thickening secondary cell wall of tracheary elements and fibers during development? From the literature, it is known that lignification follows polysaccharide synthesis. Polysaccharides are produced first, making cell wall thick, and after polysaccharide synthesis has proceeded to a certain stage, lignification initiates farthest away from the membrane (in cell corners and the middle lamella), and the proceeds inwards, but always behind polysaccharide synthesis. If an oxidative enzyme, e.g. peroxidase, is located next to plasma membrane, it could participate only when lignification is almost finished, and not in the main lignification of the thick secondary wall. Similarly, if NADPH oxidase is the only source of apoplastic H₂O₂, could that diffuse around the apoplast to the site where that is needed without being reacted closer to the site of its generation? In addition to NADPH oxidase, there are other potential enzymatic sources for apoplastic H₂O₂ (e.g. peroxidases, several types of oxidases, e.g. Kärkönen and Kuchitsu, 2015, *Phytochemistry*). Some of these could equally well be involved in H₂O₂ production needed for lignification (even in addition to NADPH oxidase) – with more broad distribution in the wall.

Response 2. We do not disagree with any of these comments. Figure 5 does indeed show the initiation of lignification in the cell corners (in both the pictures and the model). We have added sentences to the text to point out the sequence of events in developmentally-controlled lignification in xylem and fiber cells, and cited the possibility of other sources of hydrogen peroxide. We have also added a caveat to the legend to Figure 5.

I. 408-409. There are several older papers on trees that lignification is initiated in cell corners and middle lamella – please, cite some older papers. It is interesting to notice that ectopic lignin formation is also initiated in cell corners. It would be good to have a figure of developmental lignification next to this in Fig. 5 (or citation to a figure showing this).

Response 3. The legend to Figure 5 has been supplemented to address this point. We don't feel that adding an extra Figure would be helpful, as we have defined the scope of the article in the introduction. Furthermore, we believe we have been careful to not over-extend the potential applicability of the models we have presented. We believe that the following section of the text makes this clear "It has been proposed that monolignols are exported out of the cell into all areas of the apoplast where they are highly mobile [99]. However, there may be different mechanisms of monolignol transport in living tissues such as the Casparian strip (CS) versus terminally dying cell types (e.g xylem elements) during development, and in plant cells responding defensively to pathogen attack (e.g. programmed cell death and cooperative lignification in living cells (reviewed in [100])). We deliberately avoid making these

distinctions in this article, and present more generic models. We make no claims as to whether these models are correct in all details but hope that they will stimulate further experimentation”.

I. 331-332 and Figure 4 legend: “Hydrogen peroxide is generated in nearly all intracellular compartments, with a major source being the plasma membrane-associated NADPH oxidase (NOX) with its catalytic site in the apoplast.”

NADPH oxidases may be a major source for apoplastic ROS production, but since ROS are generated e.g. in chloroplasts and peroxisomes, NADPH oxidase may not be the main cellular source for H₂O₂ generation in photosynthesizing cells. NADPH oxidases transfer electrons across the plasma membrane from cytoplasmic NADPH to molecular oxygen to produce superoxide in the apoplastic side of the membrane. Thus, product generation occurs in the apoplast, but electrons originate from NADPH that is in the cytoplasmic side, and are transferred via the heme groups in the transmembrane areas (Sagi and Fluhr 2006). Hence, catalytic site locates in the cytoplasmic side and through the membrane, and product is generated in the apoplastic side.

Response 4. We agree, and have modified the text and the legend to Figure 4 to indicate that any model linking apoplastic hydrogen peroxide production to monolignol formation via C3H/APX and/or CSE/LPL2 will be difficult to prove because of the alternative generation of hydrogen peroxide in organelles within the cytosol. We have also modified Figure 4 to include the generation of hydrogen peroxide in organelles within the cytosol. We did mention the production of hydrogen peroxide in nearly all intracellular compartments, but did not specify further. We also now cite the review by Waszczak et al. Thanks for drawing our attention to it.

I. 329-344. This chapter discusses the need to transport H₂O₂ from the apoplast to cytosol to activate lysophospholipase C/CSE. Authors need to get familiar with papers about cytosolic ROS generation and antioxidants. There are multiple sources of cytoplasmic H₂O₂ generation, and efficient antioxidant systems (additionally to APX) that keep the redox balance of the cytosol highly reduced (e.g. Noctor and Foyer, 2016, Plant Cell Environm; Smirnov and Arnaud, 2019, New Phytol). However, ROS signalling between different organs and compartments is known to be present (e.g. Waszczak et al., 2018, Annu Rev Plant Biol). It has to be kept in mind that there needs to be a concentration gradient between the sides of plasma membrane with higher concentration in the apoplast so that diffusion inwards is occurring. This gradient probably occurs in the defense response with RBOHs activated. Is the gradient enough during developmental lignification when H₂O₂ is consumed by peroxidases in the apoplastic space? Please, consider this knowledge and rewrite this chapter.

Response 5. See Response 4 above. We have also now referred to the need for a concentration gradient of hydrogen peroxide across the plasma membrane to facilitate hydrogen peroxide acting as a signal to link events in the cell wall with biosynthesis in the cytosol (in both main text and legend to Figure 4).

It is not uniform how plant species' names are written (sometimes only in English, sometimes only in latin). Please, uniform the naming.

Response 6. We have checked through the manuscript, and now give each plant its common name first (with full Latin binomial), and then refer to the plant subsequently by its abbreviated binomial. e.g. thale cress (*Arabidopsis thaliana*); *A. thaliana*.

Minor comments:

I. 26 ... highly controlled polymerization procedure... The idea of “random polymerization” is not equal to highly controlled procedure – please rephrase.

Response 7. This sentence in the summary was basically correct but, we agree, a bit misleading. “At first believed to be the product of a highly controlled polymerization procedure involving just three potential monomeric components (monolignols), it is becoming increasingly clear that the composition of lignin is quite flexible”. The use of “at first” is the problem. Around 10 years ago, it was proposed by the group of Norman Lewis that lignification was more highly controlled than now believed, and this set off a vigorous debate until it was shown that alternative monomers could indeed become lignin subunits. So strictly “at first” it was proposed to be random. We have changed the sentence to start “At one point believed to be....”

I. 33-34: H₂O₂ is an oxidant for peroxidases. Laccases that also oxidise monolignols to radicals use molecular oxygen. Please, rephrase.

Response 8. We have now used the term “...one of the oxidants...” to describe hydrogen peroxide. We also now include a statement later in the review to point out that “the oxidative polymerization of monolignols is catalyzed by laccases (using molecular oxygen) and peroxidases (using hydrogen peroxide)”.

I. 65. Extracellular – Please change to “apoplatic” to be uniform with the rest of the text.

Response 9. Done.

I. 279: .. the others being CSE and CCR. – please add references in this sentence.

Response 10. The cases of CSE and CCR are discussed and referenced in the subsequent paragraphs, so we have just added“(see below)”.

I. 343: ... in the cell walls of knock-out lines of superoxide dismutase.

Response 11. Done.

I. 385: CS – write open when first mentioned.

Response 12. The full term and abbreviation are now introduced at first mention.

I. 905-906: coniferyl alcohol (G). The order of the compounds in the figure legend could be the same than that in the figure (starting with H).

Response 13. Agree that this is better. Have re-arranged the order in the Figure legend.

I. 956: (“oxidation/polymerization via peroxidases”): Peroxidase oxidizes monolignol to monolignol radical – polymerization does not require H₂O₂ or the peroxidase enzyme anymore -> remove “polymerization”.

Response 14. Correct. Changed as requested.

Fig. 1a: coloured (red) lines are not very visible.

Response 15. We have increased the thickness of the lines to make them clearer

Fig. 2: 5HG-unit (to be uniform with the text)

Response 16. Done